# Predicting the Mortality of ICU Patients by Topic Model with Machine-Learning Techniques

**DOI:** 10.3390/healthcare10061087

**Published:** 2022-06-11

**Authors:** Chih-Chou Chiu, Chung-Min Wu, Te-Nien Chien, Ling-Jing Kao, Jiantai Timothy Qiu

**Affiliations:** 1Department of Business Management, National Taipei University of Technology, Taipei 106, Taiwan; chih3c@ntut.edu.tw (C.-C.C.); cmwu@ntut.edu.tw (C.-M.W.); lingjingkao@ntut.edu.tw (L.-J.K.); 2College of Management, National Taipei University of Technology, Taipei 106, Taiwan; 3Department of Obstetrics and Gynecology, Taipei Medical University Hospital, Taipei 110, Taiwan; jtqiu1010@tmu.edu.tw; 4College of Medicine, Taipei Medical University, Taipei 110, Taiwan

**Keywords:** electronic health records, topic model, latent dirichlet allocation, machine learning, intensive care units

## Abstract

Predicting clinical patients’ vital signs is a leading critical issue in intensive care units (ICUs) related studies. Early prediction of the mortality of ICU patients can reduce the overall mortality and cost of complication treatment. Some studies have predicted mortality based on electronic health record (EHR) data by using machine learning models. However, the semi-structured data (i.e., patients’ diagnosis data and inspection reports) is rarely used in these models. This study utilized data from the Medical Information Mart for Intensive Care III. We used a Latent Dirichlet Allocation (LDA) model to classify text in the semi-structured data of some particular topics and established and compared the classification and regression trees (CART), logistic regression (LR), multivariate adaptive regression splines (MARS), random forest (RF), and gradient boosting (GB). A total of 46,520 ICU Patients were included, with 11.5% mortality in the Medical Information Mart for Intensive Care III group. Our results revealed that the semi-structured data (diagnosis data and inspection reports) of ICU patients contain useful information that can assist clinical doctors in making critical clinical decisions. In addition, in our comparison of five machine learning models (CART, LR, MARS, RF, and GB), the GB model showed the best performance with the highest area under the receiver operating characteristic curve (AUROC) (0.9280), specificity (93.16%), and sensitivity (83.25%). The RF, LR, and MARS models showed better performance (AUROC are 0.9096, 0.8987, and 0.8935, respectively) than the CART (0.8511). The GB model showed better performance than other machine learning models (CART, LR, MARS, and RF) in predicting the mortality of patients in the intensive care unit. The analysis results could be used to develop a clinically useful decision support system.

## 1. Introduction

The spread of the COVID-19 pandemic and the increasing number of infected patients are challenging global medical units, especially intensive care units (ICUs). Hospitals need to make reasonable and accurate decisions, such as how they allocate their equipment and labor, making comprehensive assessments of information and resources available at ICU. The World Health Organization advocated that hospitals regularly monitor the specific clinical variables of hospitalized patients with COVID-19 and, when feasible, analyze the variables by using medical technology [1]. However, because patients’ illnesses are rapidly changing, making quick and accurate decisions without sufficient up-to-date information is challenging for clinicians [2]. Electronic health records (EHR) are personal health electronic records that include medical records, electrocardiograms, and medical images. Researchers can analyze their archived medical information to help clinicians make critical clinical decisions [3]. Archiving personal health records electronically not only elevates hospitals’ management and service levels but also provides medical researchers with more resources that enable them to conduct related research, such as developing and verifying prediction models [4,5,6]. Using EHR data to make clinical predictions (e.g., predicting patients’ mortality, hospital stay, disease diagnoses, and onset time) is crucial in intensive care research. In other words, identifying how to effectively predict ICU patient mortality by using EHR data allows medical personnel to accurately assess the patients’ mortality risks, detect high-risk groups early, and implement interventions promptly, improving patient prognoses and enhancing care planning and resource allocation [7].

Although many studies have used EHR data, most of them have only used quantitative EHR data [8,9,10,11]. In fact, 80% of EHR data comprises semi-structured data such as patients’ physiological conditions (free-text notes and clinician progress notes) at the time of their visits [12]. To deal with huge data volumes in the form of unstructured text has become one of the main challenges for healthcare analytics. In this respect, the application of natural language processing (NLP) has received increasing attention in the medical field to bring more benefits to health organizations in a wide range of applications. The power of NLP lies in extracting information from unstructured textual data in order to form and explore new facts or hypotheses [13]. The potentials of NLP techniques, such as Latent Dirichlet Allocation (LDA) and Bidirectional Encoder Representations for Transformers (BERT), have been constantly discussed in the healthcare literature (e.g., [14,15,16,17,18]).

To maximize the use of the semi-structured EHR data, this study used the Latent Dirichlet Allocation (LDA) to build topic models. Latent Dirichlet Allocation (LDA) is a topic generation model and uses Bayes’ rule approach to treat all text modeling as a mixture of topics and vocabularies. The word “mixture” here refers to a set of elements (i.e., topics or vocabularies) that have certain probabilities of being selected. A body of a document will incorporate multiple themes, and the topics will be fluid in nature. Each document can be represented by a vector of topic probabilities, and each topic can be represented by a vector of word probabilities [19]. Many recent studies have used LDA topic modeling [20,21,22,23]. In this study, we first applied LDA to classify text in the semi-structured data to some particular topics. Subsequently, we employed five machine learning approaches, classification and regression trees (CART), logistic regression (LR), multivariate adaptive regression splines (MARS), random forest (RF), and gradient boosting (GB), to predict ICU patient mortality. From these methods, the MARS approach was selected as the benchmark for model comparison because MARS is a mature learning technology that has the advantages of high learning efficiency and strong generalizability [24,25], and CART/RF, which were developed by Breiman [26], has been widely used [27,28,29,30,31]. Regarding the advantages and disadvantages of certain machine learning methods, please refer to [32,33,34,35] for more detailed explanations.

According to the comparison results, the topics generated by LDA do contain useful information that can considerably affect the prediction accuracy of constructed models. Moreover, in our comparison of five machine learning models (CART, LR, MARS, RF, and GB), the GB model showed the best performance with the highest area under the receiver operating characteristic curve, recall, accuracy, and F1-statistic.

This study contributed to variable generation and mortality prediction through the following three aspects. First, LDA was applied to analyze the semi-structured data (i.e., patients’ diagnosis data and inspection reports) and generate some particular topics variables. The results can be used as a reference when selecting the appropriate predictors for mortality prediction. Second, the significance of the generated variables can be further analyzed by machine learning approaches, such as gradient boosting, to understand the effect of input variables at different situations on the construction of the mortality prediction model. Third, with the topic variables generated using the LDA approach, the machine learning model provides a higher AUROC value, signifying that models built using the semi-structured data more accurately predicted whether the ICU patients would die. All these results may enhance healthcare personnel’s predictions of patients’ mortality, providing patients, their families, and healthcare personnel with more information for clinical decision-making.

## 2. Materials and Methods

The flow chart of the proposed approach is illustrated in Figure 1. The data used in this study consists of structured EHR data and clinical notes data. To obtain a meaningful dataset, a list of query and data preprocessing were executed. Subsequently, the preprocessed data were input into the five machine learning models for mortality prediction. Finally, the prediction performance was evaluated by five different metrics. A detailed description of the main research procedures is presented as follows:

### 2.1. The Medical Information Mart for Intensive Car—III Dataset

The data used in this work were obtained from the Medical Information Mart for Intensive Care (MIMIC III) clinical database. MIMIC-III contained the comprehensive clinical data of patients hospitalized at the Beth Israel Deaconess Medical Center (BIDMC) in Boston, Massachusetts [36]. MIMIC-III data contained different ICU data from 2001 to 2012, where the data included patients’ vital signs, medications, data measured in labs, and observation records. Table 1 provides a breakdown of the adult population by care unit. According to the table, 49,785 hospitalization-related data were collected from 38,597 adult patients in the MIMIC-III database. Among the patients, 55.9% were males, and the median age was 65.8. The median length of an ICU stay is 2.1 days, and the median length of a hospital stay is 6.9 days.

This study received approval from the institutional review boards of the BIDMC and the MIT (Cambridge, MA, USA) to use the MIMIC-III database to perform analyses. The MIMIC-III dataset analyzed was obtained from the MIT Laboratory for Computational Physiology and a research team with which it collaborated. The dataset website URL was MIMIC-III Clinical Database [36]. Available online: https://physionet.org/content/mimiciii/1.4/ (accessed on 1 June 2021).

### 2.2. Latent Dirichlet Allocation

Latent Dirichlet Allocation (LDA) is a topic modeling algorithm for discovering the underlying topics in corpora in an unsupervised manner. Proposed by Blei et al. [19], LDA is a typical “bag of words” model that treats each text as a vocabulary frequency vector and as a collection of multiple sets of vocabularies. In addition, each group of vocabularies represents a topic, and text topics are extracted without considering the order of and relevance between the vocabularies [37,38]. Normally, an LDA builds its topic generation model through the following steps: (1) a topic is selected from the various topics in a text; (2) a vocabulary is chosen from the list of vocabularies corresponding to the topic selected; and (3) the process is repeated until all of the vocabulary in the text has been selected. Because each text consists of multiple topics that contain multiple key vocabularies, identical vocabularies may be found in different topics. Assuming that M is the number of text, K is the number of topics, Zmn is the number of times that vocabulary n appears in the different topics of Text m (where Zmn has a multinomial distribution); θm is the probability that each of the k number of topics occurs in Text m, where Dirichlet distribution (which has a hyperparameter α) is used as a priori distribution; Wmn is the nth vocabulary in Text m and has a multinomial distribution; and Φk is the probability that each vocabulary in the kth topic occurs where Dirichlet distribution (which has a hyperparameter β) is used as a priori distribution, then an overall LDA framework depicted in Figure 2 can be obtained.

We adopted the approximation algorithm proposed by [39] to filter out duplicate or incorrect notes found in the NOTEVENTS table in the MIMIC-III database, deleted unrecognizable sentences, and reserved only sentences composed of letters for text marking. Next, we used the LDA method to generate basic “topics” viewed as input variables to construct models for predicting patient mortality. Additionally, this study referenced the Grid Search method proposed by Teng et al. [40] to determine the optimal number of topics and the final LDA model to be used. According to the analysis results, 10 topics produced the optimal prediction results. Appendix A and Appendix B list the 10 topics and their corresponding keywords generated by applying the LDA method in this study.

### 2.3. Data Preprocessing

#### 2.3.1. Data Extraction

To ensure the generalizability of the analysis results, this study analyzed all patients as opposed to patients with specific diseases. Additionally, to enable one to compare the results of this study with those of relevant studies, this study set adult ICU patients older than 16 years of age and who were admitted to ICUs for the first time as its participants. The analysis data were mostly data of said patients 12 and 24 h after they were admitted to ICUs [2,41,42,43]. Figure 3 shows the detailed process of data extraction.

Table 2 provides the demographic information of the selected patient cohort after data preprocessing in our study. Among the patients, 24,252 had an ICU stay of 12 h, while 27,809 had an ICU stay of 24 h. The average age of the patients was 63 years and 56% were male. More than 70% of the patients were white, and over 80% of the patients were admitted to the ICUs because of emergencies. As many as close to 40% of patients were admitted to Medical ICUs. The patients stayed at the hospitals and ICUs for an average of 8.9 and 4.2 days, respectively.

#### 2.3.2. Variable Selection

To determine the variables to be used, we referenced relevant studies [7,41,42,44] and manually selected 16 quantitative variables based on their clinical importance in the domain from admission, chartevents, labevents, and output events data tables in the MIMIC-III dataset. The variables were Glasgow Coma Scale, heart rate, systolic blood pressure, temperature, FiO_2_, urine output, PO_2_, blood urea nitrogen, white blood cell count, potassium level, sodium level, serum bicarbonate level, bilirubin, admission type, patient’s sex, and age. Subsequently, this study adopted the data preprocessing method introduced by Guo et al. [7] to initiate a three-stage missing value processing. First, patients missing a value more than 30% were eliminated. Second, predictors missing a value more than 40% were eliminated. Third, the statistics for which the missing data rate was greater than 20% under these indicators were eliminated. The mean interpolation to interpolate the remaining missing value was then used. Then, the Information Gain Technique (Entropy) [44] was used to evaluate the importance of these 16 variables. Finally, our variable selection was based on the highest ranked attributes that scored 0.01 or more. Appendix C shows the significance rank of these features where the White Blood Cells Court is ranked the highest in the list while gender is ranked the least.

In addition to selecting the abovementioned quantitative variables, this study also extracted topic modeling variables from NOTEVENTS data. NOTEVENTS referred to clinical notes taken by doctors, nurses, imaging professionals, nutritionists, and physical therapists on patients. In the MIMIC-III database, the NOTEVENTS file contained 2,083,180 pieces of data, of which roughly 56% were data recorded by doctors or nurses and 39% were echo reports, ECG reports, and radiology reports.

#### 2.3.3. Dealing with Imbalanced Dataset

Table 3 presents the descriptive statistics of the data used in this study. The table shows that patients’ survival-to-death ratios are significantly imbalanced for both 12 and 24 h after hospital admission. Because imbalanced datasets frequently result in inaccurate model prediction results [45], researchers often balance data by adding minority samples or deleting majority samples [46]. In this study, because the sample size of ICU patients who died was much smaller than that of ICU patients who survived, we used the synthetic minority oversampling technique (SMOTE) to increase the sample size of ICU patients who died to achieve balanced results [46]. The SMOTE technique is a type of oversampling method that has been widely used in machine learning with imbalance data [47,48]. The SMOTE technique randomly generates new samples of the minority class from the nearest neighbor of the line connecting samples of the minority class. These new samples are generated based on the features of the original dataset so that they become similar to the original instances of the minority class [49].

In our study, we applied the SMOTE techniques with different percentages for different cases. As a result, several new training datasets were generated (Table 4). Take the dataset of patients 12 h after hospital admission as an example. SMOTE (900%) increased the sample with class “died” from 2384 instances to 21,456 instances. This made an incremental increase in the minority class from 9.83% in the original dataset to 49.52% in the SMOTE with 900% dataset.

#### 2.3.4. Model Validation

To validate the model’s performance after training, we used the K-fold cross-validation method [50] in this study. Using the k-fold method, we first divide the dataset into k parts, and each part will have instances of the same size. The training process is applied on all parts except one part for testing. This process is iterative and is repeated by the specified K number, where each part has the chance to be tested once. The final performance measure will be the average of all the tests’ performance of all parts. The advantage of this approach is that all instances of the entire dataset are trained and tested, so that lower variance occurs in the ensemble estimator. This ensures that the true rate estimator’s predictions are more accurate and less biased; however, this approach is computationally expensive and validation takes a long time to complete. In our study, we employed 10-fold cross-validation to construct models, which has been used in several health care and medical related studies [51,52].

### 2.4. Mortality Prediction

We assessed the effects of combining ICU patients’ structured data (vital signs and laboratory test results) and semi-structured data (diagnosis data and inspection reports) 12 and 24 h after they had been hospitalized on their mortality predictions for different periods, where death and alive were defined as “1” and “0”, respectively. ICU patient mortality-related definitions are summarized as follows:In-hospital mortality: Whether the ICU patient died during hospitalization.Short-term mortality: Whether the ICU patient died within 48 h or 72 h of hospital admission.Long-term mortality: Whether the ICU patient died within 30 days or 1 year of hospital admission.

### 2.5. Machine Learning

To illustrate the effects of adding textual data such as clinical notes and pathology reports on patient mortality predictions, this study used five classic machine learning classification algorithms to construct models for predicting ICU patient mortality. These are classification and regression trees (CART), logistic regression (LR), multivariate adaptive regression splines (MARS), random forest (RF), and gradient boosting (GB). All data mining tasks of this research were performed using python programming. Detailed descriptions of the machine learning classification algorithms are organized as follows:Classification and Regression Tree (CART)

CART is a decision tree algorithm that uses binary splitting to analyze gargantuan datasets. Through a recursive process, CART divides existing training samples into several known categories according to its predictor variables and their corresponding indicators. The training sample division process is subsequently set as a series of rules [53,54].

Logistic Regression (LR)

LR is a log probability model that can assess statistical interactions and control multivariate confidence intervals. It is most commonly used to check the risk relationships between diseases and exposures [55,56]. This study employed Python’s scikit-learn library to realize LR and selected the stochastic average gradient linear convergence algorithm as the hyperparameter setting optimization method. LR is a gradient descent method that is especially effective when the number of sample data is large.

Multivariate Adaptive Regression Splines (MARS)

MARS is a multivariate, nonparametric regression technique and a tool that accumulates several basis functions to explain nonlinear states [57]. Once objective variables are set and a set that contains selectable predictor variables is given, MARS can automate the entire model construction process, including separating meaningful and less appropriate variables, determining the interactions between predictor variables, dealing with the missing value problem by using variable clustering techniques, and avoiding overfitting by using numerous self-tests [38,58].

Random Forest (RF)

RF is an ensemble algorithm that uses decision trees as its basic classifiers [59]. It boasts the characteristic of providing accurate prediction results without having to thoroughly adjust model hyperparameters [60]. The only parameters that require thorough adjustments are the depths and number of decision trees. This study found that setting the maximum tree depth and number at 35 and 200, respectively, produced the optimal ICU patient mortality prediction results.

Gradient Boosting (GB)

GB is an ensemble learning algorithm that can be used to elevate the accuracy of different types of prediction models. It uses the negative gradient information of the loss function in the model to train models with unfavorable prediction accuracy and cumulatively integrates trained results into existing models [61,62]. This study used the scikit-learn library to realize GB, set the maximum number of iterations to 100, and trained hyperparameters by using the default values provided by the scikit-learn library.

### 2.6. Evaluation Metrics

To fully compare the effects of integrated structured and semi-structured data on ICU patient mortality predictions, this study selected five indicators (i.e., AUROC, specificity, sensitivity, precision, and F1-statistic) as the assessment tools for constructing models. Table 5 shows the confusion matrix. Detailed definitions of each assessment indicator are as follows:



(1)
$$\mathrm{Specificity}=\mathrm{TNR}=\frac{\mathrm{TN}}{\mathrm{TN}+\mathrm{FP}}$$


(2)
$$\mathrm{Sensitivity}=\mathrm{TPR}=\frac{\mathrm{TP}}{\mathrm{TP}+\mathrm{FN}}$$


(3)
$$\mathrm{Precision}=\mathrm{PPV}=\frac{\mathrm{TP}}{\mathrm{TP}+\mathrm{FP}}$$


(4)
$$F1-\mathrm{Statistic}=\frac{2\times \mathrm{Precision}\times \mathrm{Recall}}{\mathrm{Precision}+\mathrm{Recall}}$$



Specificity: The percentage of negative samples that were predicted to be negative.Sensitivity: The percentage of positive samples that were predicted to be positive.Precision: The percentage of samples that were predicted to be positive among samples that were categorized as being positive.F1-Statistic: The harmonic mean between precision and sensitivity.AUROC: The area under the receiver operating curve is primarily used to measure the classification threshold performance of classifiers. ROC is a curve consisting of points generated by the true positive rate (TPR) and false positive rate (FPR) of model. TPR signify the probabilities that models can correctly locate positive samples. Such probabilities are commonly referred to as recall rates and represent revenue. By contrast, FPR signify the probabilities that models incorrectly locate positive samples and represent losses. AUROC values range from 0 to 1, where the larger the value, the more superior the result.

## 3. Results

This study employed 10-fold cross-validation to construct models. The models constructed were using ICU patients 12 and 24 h after hospital admission to predict ICU patient mortality for different periods, including in-hospital mortality, short-term mortality, and long-term mortality. The results are presented in Table 6 and Table 7. Figure 4 and Figure 5 compare the AUROC predicted using the five machine learning methods.

### 3.1. Variable Importance

One advantage of using the GB method is that once prediction models have been built, the importance of their variables can be obtained by sorting the variable importance scores. In general, the importance score denotes the degree to which an input variable increases the value of the decision trees in the model; the more frequently that the variable is used in the decision tree, the higher its relative importance. For the GB method, the importance of an input variable is calculated using the degree to which the variable can increase the value of the decision tree at the decision tree split point multiplied by the number of samples (weights) at the node. Common decision tree value measurement methods include the Gini index, cross-entropy, and information gain (in this study, the Gini index was used to measure increases in decision tree values). For more information on how the GB method calculates the importance of input variables, please refer to Hastie et al. [63]. The variable importance obtained for the best GB model in ICU patients 12 and 24 h after hospital admission is presented in Table 8.

As presented in Table 8, when using models (constructed using the data of ICU patients 12 h after they have been admitted to hospitals) to predict in-hospital and short-term ICU patient mortality, blood urea nitrogen (x_5_) is a critical variable. By contrast, when predicting long-term ICU patient mortality (i.e., mortality within 1 year), serum bicarbonate level (x_9_), and intracranial hemorrhage (TOPIC_A3_) are more critical. When predicting in-hospital, short-term, and long-term ICU patient mortality, age (x_12_) and the Glasgow Coma Scale (x_1_) are critical variables. As for using models (constructed using the data of ICU patients 24 h after they had been admitted to hospitals) to predict in-hospital ICU patient mortality, the Glasgow Coma Scale (x_1_), blood urea nitrogen (x_5_), and serum bicarbonate level (x_9_) are critical. When predicting short-term and long-term ICU patient mortality, potassium level (x_7_) and admission type (x_10_) are more critical. When predicting in-hospital, short-term, and long-term ICU patient mortality, age (x_12_) and hydroperitoneum (TOPIC_B1_) are critical variables.

### 3.2. Prediction with Semi-Structure Data vs. Prediction w/o Semi-Structure Data

To illustrate the effects of semi-structured data on ICU patient mortality predictions, this study used the GB method to compare the ICU patient mortality prediction results obtained by models that used the structured and semi-structured data of patients 24 h after hospital admission and those obtained by models that used merely the structured data of patients 24 h after hospital admission. Table 9 shows the prediction results.

According to the table, models that used both the structured and semi-structured data of patients 24 h after hospital admission generated more accurate in-hospital, short-term, and long-term ICU patient mortality prediction results than those generated by models that used only the structured data of ICU patients. These results indicated that semi-structured data (i.e., clinical notes, which contained patients’ diagnosis data and inspection reports) contain useful information that can considerably affect the prediction accuracy of constructed models. Overall, models predicted short-term mortality more accurately than they did in-hospital and long-term mortality, and short-term, 48 h mortality predictions produced the highest AUROC values, signifying that models built using the structured and semi-structured data of ICU patients 24 h after hospital admission more accurately predicted whether the ICU patients would die 48 h after hospital admission.

## 4. Discussion

### 4.1. Principal Findings

The purpose of this study was to use a machine learning model to evaluate the impact of integrating the structured data (vital signs and laboratory test results) and semi-structured data (diagnosis data and inspection reports) collected by ICU patients during hospitalization on predicting whether ICU patients die (death = 1, survival = 0) in different periods. On the basis of the analysis results, this study presented the following findings:A longer ICU patient hospital stay signified more accumulated medical records. The increased number of medical records elevated the ICU patient mortality prediction accuracy. This study collected and used the data of ICU patients 12 and 24 h after hospital admission to construct and analyze patient mortality prediction models. Related analysis results revealed that, in general, models built using 24 h data outperformed those built using 12 h data in terms of model prediction accuracy.Overall, the prediction models predicted short-term mortality more accurately than they did long-term mortality and predicted short-term, 48 h mortality more accurately than they did all other periods. As the prediction time increased, the prediction accuracy substantially decreased. This signified that the prediction models are more suitable for short-term mortality predictions. Future studies should strengthen the long-term mortality prediction accuracy of these prediction models by increasing the duration of the data collection process of inpatients and including more factors of influence in the models.Models constructed using the five machine learning classification algorithms produced an ICU patient short-term and long-term mortality prediction accuracy of over 70%. Moreover, GB outperformed the other machine learning algorithms for all periods. These results demonstrated the rapid development of machine learning algorithms and that they can provide immense help to clinical doctors when making clinical decisions.Combining the structured and semi-structured data of ICU patients can strengthen the ICU patient mortality prediction accuracy for different periods. This confirmed that ICU patients’ clinical notes (e.g., diagnosis data and inspection reports) contain useful information that can help clinical doctors make crucial clinical decisions [64,65].Analyses on ICU patients’ semi-structured data (e.g., clinical notes and pathology reports) performed with the LDA method revealed some critical information. According to Table 7, when patients’ hospitalization data are limited (i.e., only their 12 h data are available), those who wish to predict the patients’ short-term mortality should pay attention to variables such as the patents’ ages (x_12_), Glasgow Coma Scales (x_1_), and blood urea nitrogen (x_5_); and those who wish to predict the patients’ long-term mortality should pay attention to variables such as the patents’ ages (x_12_), Glasgow Coma Scales (x_1_), serum bicarbonate levels (x_9_), and intracranial hemorrhage (TOPIC_A3_). By contrast, when patients’ hospitalization data are sufficient (i.e., their 24 h data are available), those who wish to predict the patients’ in-hospital mortality should pay attention to variables such as the patients’ ages (x_12_), hydroperitoneum (TOPIC_B1_), Glasgow Coma Scales (x_1_), blood urea nitrogen (x_5_), and serum bicarbonate levels (x_9_); and those who wish to predict the patients’ short-term and long-term mortality should pay attention to variables such as the patients’ ages (x_12_), hydroperitoneum (TOPIC_B1_), potassium levels (x_7_), and admission types (x_10_). In other words, when patients’ hospitalization data are limited, their Glasgow Coma Scales, age, blood urea nitrogen (bicarbonate), and intracranial hemorrhage will determine their prognoses. By contrast, when patients’ hospitalization data are sufficient, their Glasgow Coma Scales, age, blood urea nitrogen (bicarbonate), hydroperitoneum, and admission types will determine their prognoses. Overall, topics generated using the LDA method can extract patients’ critical medical characteristics. These medical characteristics can be used by doctors to offer personalized clinical advice according to the patients’ situations [43]. Additionally, compared with conventional methods of clustering or drawing associations with individual taxa, the LDA provides unique analytical advantages. For example, in addition to avoiding the effects of outlier samples, the LDA method can prevent the patients’ critical medical characteristics hidden in textual data from being overlooked [66].

### 4.2. Limitations

This study had a few limitations. First, because of the retrospective design, inherent biases were unavoidable. Because this study built prediction models by using the dynamic EHR data of ICU patients, the models are applicable only to patients in ICU environments or ICU-related environment. This is a common problem when constructing prediction models by using machine learning methods and dynamic EHR data [67].

Second, to collect data thoroughly and conveniently, we obtained comprehensive, dynamic patient data from a database where such data could be easily obtained. The MIMIC-III data used were obtained from BIDMC in Boston, MA, USA. Future studies should collect data from other regions and from other types of medical institutions before conducting comprehensive assessments. Moreover, because the data in this study were the medical data of ICU patients (in a large medical institution in Boston) found in the MIMIC-III database, the analysis results are not entirely applicable to ICU patients in small medical institutions. Future studies should simultaneously compare the data of ICU patients in rural medical indications and those of ICU patients in other general medical institutions to yield more comprehensive results.

Third, the medical data were only those of patients who had been admitted to ICUs for the first time and did not include those of patients who were readmitted. Because ICU patient readmissions often lead to excessive use of medical resources and higher financial risks for medical institutions, analyzing the morbidity and mortality of readmitted ICU patients will benefit patients and medical institutions more pronouncedly [68]. Future studies can collect the data of ICU patients who have been hospitalized multiple times, perform comprehensive assessments for different time series, or provide different types of analysis results to patients, health care personnel, and patients’ families to enable them to use such reference information when making related assessments.

Fourth, the LDA topic modeling method adopted in this study is an alternative nonparametric method that normally requires one to observe data complexity to determine the number of topics required. Such an analysis procedure often violates the objective analysis principle. In addition, standard LDA models frequently interpret data as disordered “bag of words” and remove them from analyses, resulting in wasteful use of information [69]. Follow-up studies that resolve these problems by conducting more comprehensive assessments and analyses will ensure more objective and complete study results.

## 5. Conclusions

In this study, in addition to using quantitative data (e.g., ICU patients’ vital signs and laboratory test results), we use an LDA method to model the semi-structured data (e.g., patients’ clinical notes and pathology reports) of ICU patients and discuss the effectiveness of combining LDA and five machine learning methods to predict ICU patient mortality. Our results revealed that the semi-structured data (diagnosis data and inspection reports) of ICU patients contain useful information that can assist clinical doctors in making critical clinical decisions. However, the prediction models built in this study are mainly used to predict ICU patient mortality, and follow-up studies are required to enable the models to make other clinical predictions, for instance, hospital stay, complication, and disease predictions.

Possible directions for follow-up studies are as follows: first, they may collect the structured and semi-structured data of patients in different departments (e.g., cardiology or nephrology departments) and with different diseases (e.g., sepsis) and perform more detailed classifications and analyses. Second, they may collect patient data from different medical departments, such as the outpatient departments, emergency departments, inpatient departments, and ICUs, and conduct more comprehensive model construction and assessments to strengthen the generalizability and applicability of the models. Third, they may collect and integrate different types of unstructured data, for instance, consultation processes in hospitals, patient demands, and messages left by patients on social media such as Facebook, Instagram, and Twitter when building models to increase model prediction accuracy. Moreover, the NLP research is gradually dominated by the use of some new transformer models (e.g BERT). Future studies may aim to combine other new topic modeling tools, such as BERT, to evaluate the ability of the proposed prediction scheme.

## Figures and Tables

**Figure 1 healthcare-10-01087-f001:**
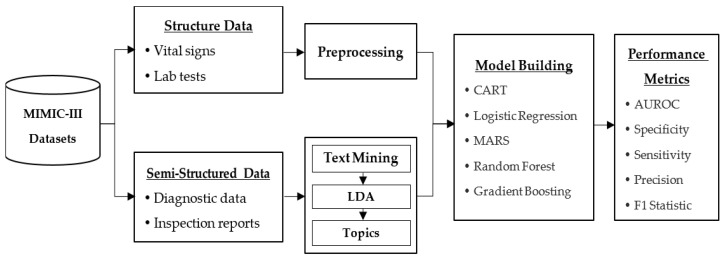
Research scheme.

**Figure 2 healthcare-10-01087-f002:**
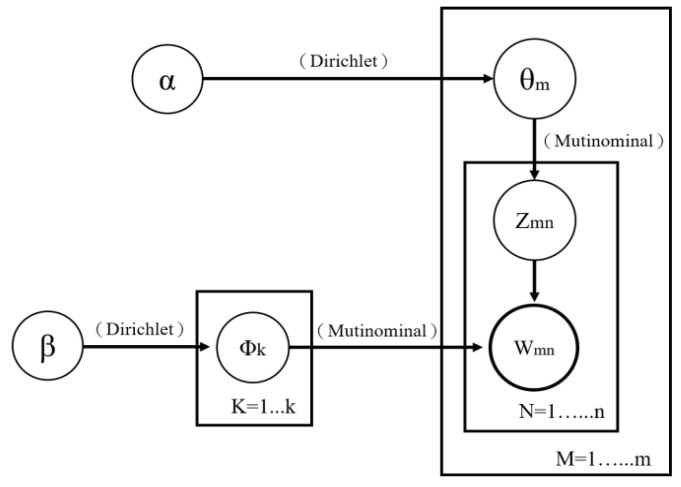
LDA model framework.

**Figure 3 healthcare-10-01087-f003:**
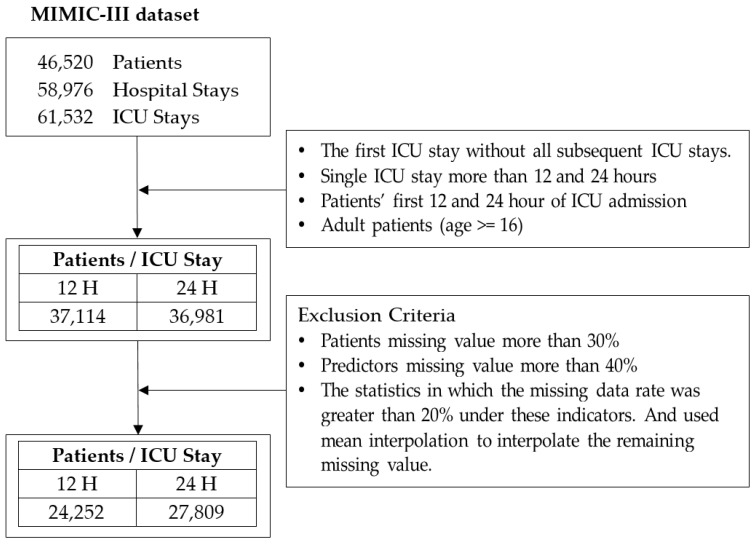
The detailed process of data extraction.

**Figure 4 healthcare-10-01087-f004:**
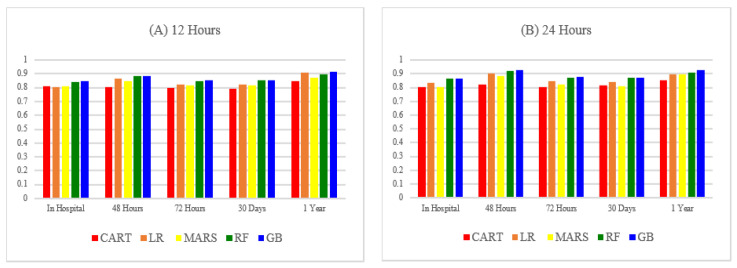
The AUROCs of different classifiers (**A**) based on 12 h dataset (**B**) based on 24 h dataset.

**Figure 5 healthcare-10-01087-f005:**
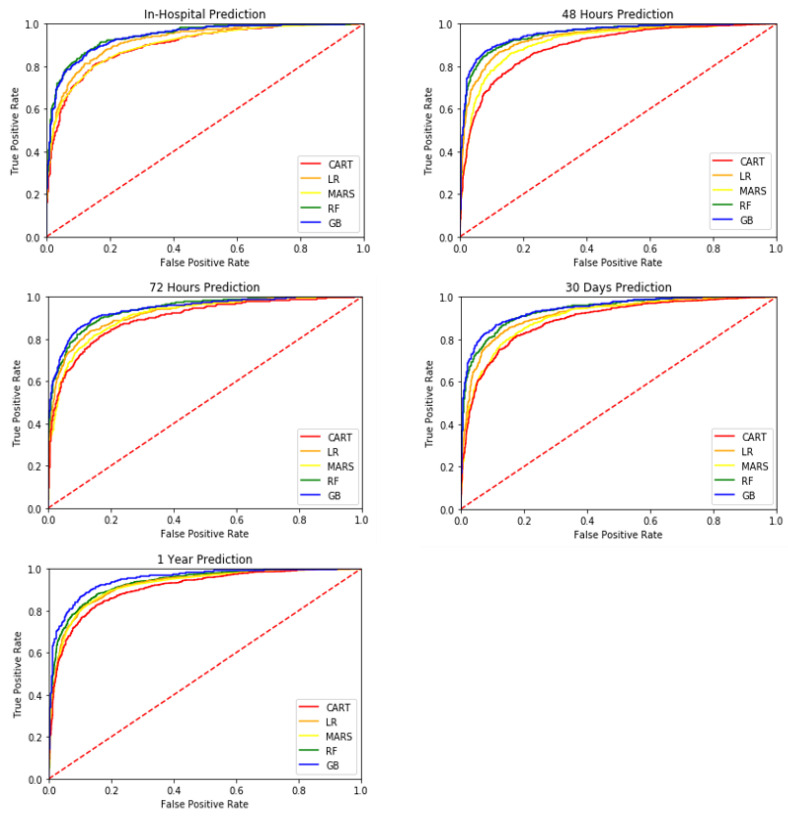
AUROC of different classifiers based on 24 h dataset.

**Table 1 healthcare-10-01087-t001:** Details of the MIMIC-III patient population for patients aged 16 years and above.

Adult Patients Critical Care Unit	Total
Distinct patients	38,597
Hospital admissions	49,785
Distinct ICU stays	53,423
Coronary Care Unit (CCU)	7726 (14.5%)
Cardiac Surgery Recovery Unit (CSRU)	9854 (18.4%)
Medical Intensive Care Unit (MICU)	21,087 (39.5%)
Surgical Intensive Care Unit (SICU)	8891 (16.6%)
Trauma Surgical Intensive Care Unit (TSICU)	5865 (11.1%)
Age, years, median [Q1–Q3]	65.8 [52.8–77.8]
Gender, male	27,983 (55.9%)
ICU length of stay, median days [Q1–Q3]	2.1 [1.2–4.6]
Hospital length of stay, median days [Q1–Q3]	6.9 [4.1–11.9]
ICU mortality	4565 (8.5%)
Hospital mortality	5748 (11.5%)
A mean of # is available for each hospital admission.	
Chartevents (330,712,483)	6642.81
Inputevents (21,136,926)	424.56
Outputevents (4,349,218)	87.36
Labevents (27,854,055)	559.49
Noteevents (2,083,180)	41.84

ICU, intensive care unit.

**Table 2 healthcare-10-01087-t002:** Selected Patient Demographic Information.

	12 h			24 h		
	Overall	Dead at Hospital	Alive at Hospital	Overall	Dead at Hospital	Alive at Hospital
General (%)						
Number	24,252 (100%)	2384 (9.83%)	21,868 (90.17%)	27,809 (100%)	2559 (9.20%)	25,250 (90.80%)
Age [Q1–Q3]	63.02 [50.96–78.16]	70.76 [61.19–83.32]	62.17 [50.07–77.27]	63.06 [51.32–77.82]	70.88 [61.43–83.32]	62.26 [50.51–76.97]
Gender (male)	13,675 (56.38%)	1267 (9.27%)	12,408 (90.73%)	15,805 (56.83%)	1353 (8.56%)	14,452 (91.44%)
Ethnicity (%)						
Asian	598 (2.47%)	56 (9.36%)	542 (90.64%)	680 (2.45%)	62 (9.12%)	618 (90.88%)
Black	1930 (7.96%)	106 (5.49%)	1824 (94.51%)	2142 (7.70%)	113 (5.28%)	2029 (94.72%)
Hispanic	841 (3.47%)	44 (5.23%)	797 (94.77%)	919 (3.30%)	49 (5.33%)	870 (94.67%)
White	17,262 (71.18%)	1604 (9.29%)	15,658 (90.71%)	19,809 (71.23%)	1733 (8.75%)	18,076 (91.25%)
Other	3621 (14.93%)	574 (15.85%)	3047 (84.15%)	4259 (15.32%)	602 (14.13%)	3657 (85.87%)
Admission Type (%)						
Urgent	562 (2.32%)	69 (12.28%)	493 (87.72%)	667 (2.40%)	77 (11.54%)	590 (88.46%)
Emergency	21,096 (86.99%)	2284 (10.83%)	18,812 (89.17%)	22,890 (82.31%)	2427 (10.60%)	20,463 (89.40%)
Elective	2594 (10.70%)	31 (1.20%)	2563 (98.80%)	4252 (15.29%)	55 (1.29%)	4197 (98.71%)
Site (%)						
MICU	9654 (39.81%)	1099 (11.38%)	8555 (88.62%)	10,309 (37.07%)	1187 (11.51%)	9122 (88.49%)
SICU	3942 (16.25%)	476 (12.08%)	3466 (87.92%)	4543 (16.34%)	501 (11.03%)	4042 (88.97%)
CCU	3925 (16.18%)	334 (8.51%)	3591 (91.49%)	4316 (15.52%)	360 (8.34%)	3956 (91.66%)
CSRU	2955 (12.18%)	126 (4.26%)	2829 (95.74%)	4482 (16.12%)	149 (3.32%)	4333 (96.68%)
TSICU	3776 (15.57%)	349 (9.24%)	3427 (90.76%)	4159 (14.96%)	362 (8.70%)	3797 (91.30%)
Outcomes						
Hospital LOS (days) [Q1–Q3]	8.98 [3.79–10.66]	9.16 [2.76–11.40]	8.97 [3.86–10.58]	8.95 [3.88–10.47]	9.27 [2.77–11.49]	8.92 [3.96–10.34]
ICU LOS (days) [Q1–Q3]	4.26 [1.37–4.57]	6.64 [2.12–8.13]	4.00 [1.24–3.97]	4.15 [1.26–4.17]	6.68 [2.08–8.12]	3.89 [1.22–3.89]
Hospital death (%)	2384 (9.83%)	-	-	2559 (9.20%)	-	-

MICU Denotes Medical ICU; SICU Denotes Surgical ICU; CCU Denotes Coronary Care Unit; CSRU Denotes Cardiac Surgery Recovery Unit; TSICU Denotes Trauma Surgical ICU.

**Table 3 healthcare-10-01087-t003:** Demographic information of the selected patient cohort.

		In-Hospital Mortality	Short-Term Mortality		Long-Term Mortality	
			48 h	72 h	30 Days	1 Year
12 h	Number of Survive	21,868	23,873	23,590	21,932	21,839
	Number of death	2384	379	662	2320	2413
	Mortality ratio	9.83%	1.56%	2.73%	9.57%	9.95%
24 h	Number of Survive	25,250	27,409	27,103	25,324	25,219
	Number of death	2559	400	706	2485	2590
	Mortality ratio	9.20%	1.44%	2.54%	8.94%	9.31%

**Table 4 healthcare-10-01087-t004:** Number of instances increased by SMOTE technique.

Hours after Hospital Admission	Mortality		Percentage of SMOTE Increase	Class “Survived”	Class “Died”
12 h	In-Hospital		900%	21,868	21,456
	Short Term	48 h	6200%	23,873	23,498
		72 h	3500%	23,590	23,170
	Long Term	30 Days	900%	21,932	20,880
		1 Year	900%	21,839	21,717
24 h	In-Hospital		900%	25,250	23,031
	Short Term	48 h	6800%	27,409	27,200
		72 h	3800%	27,103	26,828
	Long Term	30 Days	1000%	25,324	24,850
		1 Year	900%	25,219	23,310

**Table 5 healthcare-10-01087-t005:** Confusion Matrix.

		Prediction	
		Positive	Negative
Actual	Positive	True Positive (TP)	False Negative (FN)
	Negative	False Positive (FP)	True Negative (TN)

**Table 6 healthcare-10-01087-t006:** Comparisons between different models constructed using 12 h dataset in terms of their prediction accuracy.

Metric	Method	In-Hospital Mortality	Short-Term Mortality		Long-Term Mortality	
			48 h	72 h	30 Days	1 Year
AUROC	CART	0.8101	0.8033	0.8006	0.7925	0.8471
	LR	0.8029	0.8659	0.8222	0.8224	0.9082
	MARS	0.8124	0.8502	0.8195	0.8170	0.8716
	RF	0.8415	0.8867	0.8498	0.8543	0.8953
	GB	0.8489	0.8862	0.8542	0.8556	0.9171
Specificity	CART	0.7416	0.8425	0.7071	0.7544	0.8181
	LR	0.7352	0.7921	0.7455	0.7743	0.8521
	MARS	0.5696	0.6243	0.5861	0.6344	0.7090
	RF	0.1735	0.2408	0.1638	0.1900	0.3796
	GB	0.7528	0.8712	0.7578	0.7907	0.9259
Sensitivity	CART	0.7460	0.7211	0.7864	0.7553	0.8080
	LR	0.7126	0.8188	0.7457	0.7158	0.8000
	MARS	0.8599	0.8842	0.8665	0.8275	0.8480
	RF	0.9912	0.9842	0.9970	0.9933	0.9760
	GB	0.7810	0.7211	0.7864	0.7647	0.6720
Precision	CART	0.2323	0.1167	0.2179	0.2605	0.0788
	LR	0.2273	0.1041	0.2400	0.2756	0.0952
	MARS	0.1731	0.0636	0.1785	0.2059	0.0531
	RF	0.1117	0.0361	0.1101	0.1231	0.0294
	GB	0.2487	0.1391	0.2520	0.2950	0.1487
F1-Statistic	CART	0.3542	0.2009	0.3413	0.3874	0.1436
	LR	0.3447	0.1848	0.3632	0.3979	0.1702
	MARS	0.2882	0.1187	0.2960	0.3297	0.1000
	RF	0.2007	0.0696	0.1983	0.2191	0.0571
	GB	0.3773	0.2332	0.3817	0.4258	0.2435

CART, classification and regression trees; LR, logistic regression; MARS, multivariate adaptive regression splines; RF, random forest; and GB, gradient boosting.

**Table 7 healthcare-10-01087-t007:** Comparisons between different models constructed using 24 h dataset in terms of their prediction accuracy.

Metric	Method	In-Hospital Mortality	Short-Term Mortality		Long-Term Mortality	
			48 h	72 h	30 Days	1 Year
AUROC	CART	0.8049	0.8246	0.8064	0.8140	0.8511
	LR	0.8331	0.9014	0.8438	0.8434	0.8987
	MARS	0.8053	0.8843	0.8250	0.8102	0.8935
	RF	0.8623	0.9203	0.8705	0.8710	0.9096
	GB	0.8623	0.9249	0.8760	0.8736	0.9280
Specificity	CART	0.7639	0.7927	0.7537	0.7658	0.8197
	LR	0.7707	0.8313	0.7812	0.7684	0.8772
	MARS	0.6443	0.6151	0.6115	0.6054	0.6765
	RF	0.2520	0.3011	0.2056	0.2301	0.4212
	GB	0.8184	0.8507	0.8123	0.7882	0.9316
Sensitivity	CART	0.7402	0.8071	0.7578	0.7607	0.7983
	LR	0.7278	0.8090	0.7597	0.7683	0.7642
	MARS	0.8023	0.9137	0.8512	0.8409	0.8992
	RF	0.9899	0.9848	0.9932	0.9866	0.9580
	GB	0.7427	0.8325	0.7618	0.7821	0.7563
Precision	CART	0.2499	0.0905	0.2303	0.2425	0.0658
	LR	0.2658	0.1168	0.2673	0.2601	0.0955
	MARS	0.1933	0.0572	0.1756	0.1736	0.0423
	RF	0.1233	0.0348	0.1084	0.1122	0.0257
	GB	0.3030	0.1248	0.2831	0.2669	0.1495
F1-Statistic	CART	0.3736	0.1628	0.3532	0.3678	0.1216
	LR	0.3894	0.2041	0.3955	0.3887	0.1698
	MARS	0.3116	0.1077	0.2912	0.2878	0.0809
	RF	0.2193	0.0672	0.1955	0.2014	0.0500
	GB	0.4304	0.2171	0.4128	0.3980	0.2497

CART, classification and regression trees; LR, logistic regression; MARS, multivariate adaptive regression splines; RF, random forest; and GB, gradient boosting.

**Table 8 healthcare-10-01087-t008:** The selected six important variables for 12 h and 24 h datasets by using GB.

Dataset	Order of Variable Importance	In-Hospital Mortality	Short-Term Mortality		Long-Term Mortality	
			48 h	72 h	30 Days	1 Year
12 h	1	x_1_	x_1_	x_1_	x_1_	x_1_
	2	x_12_	x_5_	x_12_	x_12_	x_9_
	3	x_5_	x_12_	x_5_	x_5_	TOPIC_A3_
	4	x_2_	TOPIC_A3_	x_6_	x_9_	x_12_
	5	x_6_	x_9_	TOPIC_A3_	x_2_	x_2_
	6	x_4_	x_3_	x4	x_4_	x_4_
24 h	1	x_1_	x_7_	x_7_	x_7_	x_7_
	2	x_5_	x_10_	TOPIC_B1_	x_12_	TOPIC_B1_
	3	x_12_	TOPIC_B1_	x_12_	x_6_	x_10_
	4	x_9_	x_12_	x_10_	TOPIC_B1_	x_12_
	5	TOPIC_B1_	x_8_	x_3_	x_10_	x_3_
	6	x_2_	x_1_	x_5_	x_3_	x_1_

**Table 9 healthcare-10-01087-t009:** Comparisons (made using the GB method) of the prediction results generated by models based on 24 h dataset.

Dataset	Metric	In-Hospital Mortality	Short-Term Mortality		Long-Term Mortality	
			48 h	72 h	30 Days	1 Year
With semi-structure data	AUROC	0.8623	0.9249	0.8760	0.8736	0.9280
	Specificity	0.8184	0.8507	0.8123	0.7882	0.9316
	Sensitivity	0.7427	0.8325	0.7618	0.7821	0.7563
	Precision	0.3030	0.1248	0.2831	0.2669	0.1495
	F1-Statistic	0.4304	0.2171	0.4128	0.3980	0.2497
Without semi-structure data	AUROC	0.8545	0.9141	0.8643	0.8683	0.9152
	Specificity	0.8113	0.8276	0.8046	0.7932	0.9215
	Sensitivity	0.7389	0.8426	0.7564	0.7687	0.7143
	Precision	0.2939	0.1111	0.2735	0.2682	0.1265
	F1-Statistic	0.4205	0.1963	0.4017	0.3976	0.2149

## Data Availability

Not applicable.

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
