# Peer review of "Predicting the Mortality of ICU Patients by Topic Model with Machine-Learning Techniques"

_healthcare, 2022, doi:10.3390/healthcare10061087_

Round 1

Reviewer 1 Report

This paper discusses the use of Machine learning and NLP to predict the mortality in ICU units, which is very interesting. However, there are some issues that should be fixed before deciding the suitability of this paper to be published. Here are some comments to the authors to enhance their work.

CCIS Schedule Change Request

Course

Current timing

Suggested timing

Suggested venue

Remarks

IS321

1,3: 11-12:15

4: 11-11:50

1,2,3,4: 9-9:50

Move to GA05

SE401

1,2,3,4: 9-9:50

1,2,3,4: 11-11:50

Move to GA13

SE489

1,2,3,4: 4-4:50

1,2,3,4: 8-8:50

Move to GA-16

IS311/SE371

2,4: 2-3:50 (2 hours)

1,2,3,4:

10-10:50 (GA16)

OR

11-11:50 (GA11)

SE411

1,2,3,4: 8-8:50

1,2,3,4: 3-3:50

Move to GA-16

Author’s comments

  • Please revise the title to avoid the ambiguity. You can say Prediction the Mortality of  ICU Patients Using ….
  • The subtract should be revised by improving the problem statement and contribution. Show the significance of your work comparing with the previous works.
  • the transition between the ideas in introduction should be enhanced. I advise to rewrite the introduction
  • some statements should be proved by adding citation. For example, “Although many studies have used EHR data, most of them have used quantitative 54 EHR data. However, not all EHR data are quantitative.” Please mention some of those studies or add citations for this
  • The problem statement and contribution should be enhanced to show the significance of the work.
  • In the introduction, it will be great if the authors discuss the pros and cons of each Machine learning algorithms to conclude the importance of their work.
  • The approach should be augmented and explained. Some issues found. For example, how did the authors choose the selected variables? Did you use variable correlation or any specific technique?
  • The literature review should be expanded and augmented with recent references (last 5 years ) in NLP and Machine learning for prediction in healthcare. The initiatives of combing both on unstructured, semi-structured, and structured. The, show the research gap that you want to solve.
  • In the proposed approach, the authors should restructure the preprocessing process. You may add figures to enhance the readability.
  • The framework should be designed to highlight all the methods used. For example, the authors are advised to revisit their approach by showing the imbalance issue and machine learning algorithms. This also should be highlighted in the proposed approach figure. Did oversampling used to enhance the machine learning algorithms? Then show how. In addition, the 10-fold cross-validation to construct models.
  • All the figures’ captions should be revised to be informative and illustrative.

I will be delighted to see the updated version of this work with high quality improvements. 

Author Response

Response to Reviewer #1

We would like to thank you for your valuable recommendations that make the paper more complete and readable. We have made the following adjustments based on your suggestions.

Content Issues :

  1. Please revise the title to avoid the ambiguity. You can say Prediction the Mortality of ICU Patients Using ….

Corrections and adjustments:

Thanks for your valuable suggestions. We have changed the title of the manuscript accordingly.

  1. The abstract should be revised by improving the problem statement and contribution. Show the significance of your work comparing with the previous works.

Corrections and adjustments:

Thanks for your valuable suggestions. We have modified the phrasing in the abstract to enhance the problem statement and contribution of our work.

  1. The transition between the ideas in introduction should be enhanced. I advise to rewrite the introduction

Corrections and adjustments:

Thanks for your valuable suggestions. We have rewritten the introduction to enhance our research ideas.

  1. Some statements should be proved by adding citation. For example, “Although many studies have used EHR data, most of them only have used quantitative EHR data.” Please mention some of those studies or add citations for this

Corrections and adjustments:

Thanks for your valuable suggestions. After studying literatures, we have included some recent literature on the utilization of quantitative EHR data in the introduction section.

  1. The problem statement and contribution should be enhanced to show the significance of the work.

Corrections and adjustments:

Thanks for your valuable suggestions. We have rewritten the introduction to enhance the problem statement and contribution of our work.

  1. In the introduction, it will be great if the authors discuss the pros and cons of each Machine learning algorithms to conclude the importance of their work.

Corrections and adjustments:

Thanks for your valuable suggestions. We have included the reason why we used some of the machine learning algorithms in the introduction. We also provide literature[1] on the strengths and weaknesses of some machine learning methods for readers to refer to.

  1. The approach should be augmented and explained. Some issues found. For example, how did the authors choose the selected variables? Did you use variable correlation or any specific technique?

Corrections and adjustments:

Thanks for your valuable suggestions. Basically, we first referenced relevant studies [2-5] and manually selected 16 quantitative variables based on their clinical importance in the domain. Then, the Information Gain Technique (Entropy) was used to evaluate the importance of all these 16 variables. The variable selection in this study was based on the highest ranked attributes that scored 0.01 or more. All these description have been added in the 2.3 Data Preprocessing section.

  1. The literature review should be expanded and augmented with recent references (last 5 years ) in NLP and Machine learning for prediction in healthcare. The initiatives of combing both on unstructured, semi-structured, and structured. The, show the research gap that you want to solve.

Corrections and adjustments:

Thanks for your valuable suggestions. As we known, the Latent Dirichlet Allocation (LDA) model is certainly a successful approach that spanned a plethora of publications. However, the NLP research is gradually dominated by the use of other transformer models (e.g BERT). We have added some recent references in NLP and machine learning in the manuscript and included some description of the utilization of BERT as part of the future work.

  1. In the proposed approach, the authors should restructure the preprocessing process. You may add figures to enhance the readability.

Corrections and adjustments:

Thanks for your valuable suggestions. We have restructured the preprocessing process and added the description of data extraction, variable selection, dealing with imbalanced dataset, and model validation in the “2.3 Data Preprocessing” section to enhance the readability.

  1. The framework should be designed to highlight all the methods used. For example, the authors are advised to revisit their approach by showing the imbalance issue and machine learning algorithms. This also should be highlighted in the proposed approach figure. Did oversampling used to enhance the machine learning algorithms? Then show how. In addition, the 10-fold cross-validation to construct models.

Corrections and adjustments:

Thanks for your valuable suggestions. We have added the phrasing in the “2.3 Data Preprocessing” section to deal with imbalanced dataset issue. The employed 10-fold cross-validation was also highlighted in the “Model validation” section.

  1. All the figures’ captions should be revised to be informative and illustrative.

Corrections and adjustments:

Thanks for your valuable suggestions. All the figures’ captions have been modified to be informative and illustrative.

  1. Fatima, M. and M. Pasha, Survey of machine learning algorithms for disease diagnostic. Journal of Intelligent Learning Systems and Applications, 2017. 9(01): p. 1.
  2. Purushotham, S., et al., Benchmarking deep learning models on large healthcare datasets. Journal of Biomedical Informatics, 2018. 83: p. 112-134.
  3. Guo, C.H., M.L. Lu, and J.F. Chen, An evaluation of time series summary statistics as features for clinical prediction tasks. Bmc Medical Informatics and Decision Making, 2020. 20(1).
  4. Yu, R.X., et al., Using a Multi-Task Recurrent Neural Network With Attention Mechanisms to Predict Hospital Mortality of Patients. Ieee Journal of Biomedical and Health Informatics, 2020. 24(2): p. 486-492.
  5. Lin, Y.W., et al., Analysis and prediction of unplanned intensive care unit readmission using recurrent neural networks with long shortterm memory. Plos One, 2019. 14(7).

Reviewer 2 Report

I would like to thank the authors very much, the article was an interesting read for me. The methodology is largely clear, and the manuscript is well-written as well. I also commend the authors on considering the study limitations thoroughly. In general, I believe that the study has a very good potential. However, there are some points that should be considered in the next version, please.

(1)

The Latent Dirichlet Allocation (LDA) model is certainly a successful approach that spanned a plethora of publications. However, the NLP research is currently dominated by the use of transformer models (e.g BERT). Further, there are specialized models (e.g. BioBERT), which could be quite applicable in the present work. In the recent years, other methods (e.g. LDA, Bag of words) have been losing ground to transformers. That said, I believe that the study should touch on that point as part of the future work or limitations.

(2)

The contributions should be described in a better way in order to do justice to this work. The last paragraph of the introduction mentions that succinctly. In my view, the contributions should be emphasized with respect to the usefulness of fusing the textual features with structured data.

(3)

Likewise, the discussion should refer to studies that implemented the core idea of the methodology here, which is integrating unstructured text notes with standard data. For example:

https://doi.org/10.1109/BigData50022.2020.9378073

(4)

In line 226, it is mentioned that:

We filtered out duplicate or incorrect notes found in the NOTEVENTS table in the 226 MIMIC-III database, deleted unrecognizable sentences, and reserved only sentences com-227 posed of letters for text marking.

Please could you elaborate further on how that filtering process was implemented. 

(5)

It is not clear how the authors used the synthetic samples generated by SMOTE. Ideally, the synthetically generated samples should be used only for enlarging the train set, and should not be included at all in the test set. This is an essential part to clarify in the methodology, please.

(6)

A few references should be cited, please.

Chawla, N. V., Bowyer, K. W., Hall, L. O., & Kegelmeyer, W. P. (2002). SMOTE: synthetic minority over-sampling technique. Journal of Artificial Intelligence Research, 16, 321-357.

Pedregosa, F., Varoquaux, G., Gramfort, A., Michel, V., Thirion, B., Grisel, O., ... & Duchesnay, E. (2011). Scikit-learn: Machine learning in Python. Journal of Machine Learning Research, 12, 2825-2830.

Breiman, L. (2001). Random forests. Machine Learning, 45(1), 5-32.

Author Response

Response to Reviewer #2

We would like to thank you for your valuable recommendations that make the paper more complete and readable. We have made the following adjustments based on your suggestions.

Content Issues :

  1. The Latent Dirichlet Allocation (LDA) model is certainly a successful approach that spanned a plethora of publications. However, the NLP research is currently dominated by the use of transformer models (e.g BERT). Further, there are specialized models (e.g. BioBERT), which could be quite applicable in the present work. In the recent years, other methods (e.g. LDA, Bag of words) have been losing ground to transformers. That said, I believe that the study should touch on that point as part of the future work or limitations.

Corrections and adjustments:

Thanks for your valuable suggestions. As the reviewer pointed out, the Latent Dirichlet Allocation (LDA) model is certainly a successful approach that spanned a plethora of publications. However, the NLP research is gradually dominated by the use of other transformer models (e.g BERT). We have added some recent references in NLP and machine learning in the manuscript and included some description of the utilization of BERT as part of the future work.

  1. The contributions should be described in a better way in order to do justice to this work. The last paragraph of the introduction mentions that succinctly. In my view, the contributions should be emphasized with respect to the usefulness of fusing the textual features with structured data.

Corrections and adjustments:

Thanks for your valuable suggestions. We have modified the structure and phrasing in the introduction to enhance the problem statement and contribution of our work. The usefulness of fusing the textual features with structured data has been emphasized.

  1. Likewise, the discussion should refer to studies that implemented the core idea of the methodology here, which is integrating unstructured text notes with standard data. For example: https://doi.org/10.1109/BigData50022.2020.9378073

Corrections and adjustments:

Thanks for your valuable suggestions. The discussion has included some studies [1, 2] that implemented the core idea of integrating unstructured text notes with standard data.

  1. In line 226, it is mentioned that: We filtered out duplicate or incorrect notes found in the NOTEVENTS table in the 226 MIMIC-III database, deleted unrecognizable sentences, and reserved only sentences com-227 posed of letters for text marking. Please could you elaborate further on how that filtering process was implemented.

Corrections and adjustments:

Thanks for your valuable suggestions. To identify and filter out duplicate or incorrect notes found in the NOTEVENTS table in the MIMIC-III database, we have adopted the approximation algorithm proposed by [3]. The algorithm consists of three phases: (1) Minhashing with Locality Sensitive Hashing [4, 5]; (2) a clustering method using tree-structured disjoint sets; and (3) classification of near-duplicates via pairwise comparison of notes in each cluster. The algorithm can be used to analyze large clinical note corpora with limited available memory space and has been described previously [6]. The detail procedure can be referred to the mentioned literature [3]

  1. It is not clear how the authors used the synthetic samples generated by SMOTE. Ideally, the synthetically generated samples should be used only for enlarging the train set, and should not be included at all in the test set. This is an essential part to clarify in the methodology, please.

Corrections and adjustments:

Thanks for your valuable suggestions. We have added the phrasing in the “2.3 Data Preprocessing” section to deal with imbalanced dataset issue. The employed 10-fold cross-validation was also highlighted in the “Model validation” section.

  1. A few references should be cited, please.

Corrections and adjustments:

Thanks for your valuable suggestions. After studying literatures, we have included some literature accordingly.

  1. Arnaud, É., et al. Deep learning to predict hospitalization at triage: Integration of structured data and unstructured text. in 2020 IEEE International Conference on Big Data (Big Data). 2020. IEEE.
  2. Zhou, S.-M., et al., Predicting Hospital Readmission for Campylobacteriosis from Electronic Health Records: A Machine Learning and Text Mining Perspective. Journal of Personalized Medicine, 2022. 12(1): p. 86.
  3. Gabriel, R.A., et al., Identifying and characterizing highly similar notes in big clinical note datasets. Journal of biomedical informatics, 2018. 82: p. 63-69.
  4. Broder, A.Z. On the resemblance and containment of documents. in Proceedings. Compression and Complexity of SEQUENCES 1997 (Cat. No. 97TB100171). 1997. IEEE.
  5. Broder, A.Z., et al., Min-wise independent permutations. Journal of Computer and System Sciences, 2000. 60(3): p. 630-659.
  6. Shenoy, S., et al., Deduplication in a massive clinical note dataset. arXiv preprint arXiv:1704.05617, 2017.

Round 2

Reviewer 1 Report

The authors did well to respond to my comments with significant improvement reflected in the manuscript. thus, I recommend accepting the paper in its form.  

Author Response

We would like to thank you for your valuable comments that make the paper more complete and readable.

Reviewer 2 Report

Thanks very much for accommodating the feedback. The new version is a substantial improvement.

However, please consider one last comment regarding the introduction. I find it important to position the introduction more properly within the context of utilizing NLP in the healthcare context. Adding a couple of sentences or a small paragraph would be good for that purpose, especially that the intro seems now (relatively) brief and short. Recent studies that discussed that aspect should be included, for example:

http://dx.doi.org/10.5220/0010414508250832

https://doi.org/10.1016/j.tacc.2021.02.007

Author Response

Thanks for your valuable suggestions. To position the utilization of NLP in the healthcare context, we have added a couple of sentences in the introduction section. For examples, “To deal with huge data volumes in the form of unstructured text has become one of the main challenges for healthcare analytics. In this respect, the application of natural language processing (NLP) has received increasing attention in the medical field to bring more benefits to health organizations in a wide range of applications.” Recent studies that discussed that aspect also included.
